# Time-resolved electrical detection
of chiral edge vortex braiding

İ. Adagideli,[1,2] F. Hassler,[3] A. Grabsch,[2] M. Pacholski,[2]
and C.W.J. Beenakker[2]

**1** Faculty of Engineering and Natural Sciences, Sabancı University,
Orhanlı-Tuzla, 34956, Turkey
**2** Instituut-Lorentz, Universiteit Leiden, P.O. Box 9506,
2300 RA Leiden, The Netherlands
**3** JARA-Institute for Quantum Information, RWTH Aachen University,
52056 Aachen, Germany

August 2019

## Abstract

A $2\pi$ phase shift across a Josephson junction in a topological superconductor injects vortices into the chiral edge modes at opposite ends of the junction. When two vortices are fused they transfer charge into a metal contact. We calculate the time dependent current profile for the fusion process, which consists of $\pm e/2$ charge pulses that flip sign if the world lines of the vortices are braided prior to the fusion. This is an electrical signature of the non-Abelian exchange of Majorana zero-modes.

# 1  Introduction

An interesting and potentially useful line of research in electronic quantum transport is to study the injection, propagation, and detection of single-electron wave packets [1, 2]. These studies are inspired by analogies with quantum optics, where a single-photon source is an elementary building block of devices. For single-particle excitations in the Fermi sea the elementary wave packet goes by the name of *leviton* [3]: A voltage pulse over a tunnel barrier of integrated amplitude equal to a flux quantum injects one electron charge, without any particle-hole excitations if the time dependence is Lorentzian [4–6].

Single-electron levitons have been realized experimentally in a two-dimensional (2D) electron gas [7, 8]. In these systems the chiral motion in quantum Hall edge channels provides for a means of propagation that is not hindered by impurity scattering [9, 10]. A leviton could function as a "flying qubit" for quantum information processing [11–13], transferring entanglement between immobile qubits in quantum dots.

Superconducting analogues of the leviton [14, 15] are of interest in the context of superconducting platforms for quantum computation. For the superconducting counterpart to the quantum Hall effect one can turn to a 2D topological superconductor, formed by the proximity effect on the surface of a 3D topological insulator [16]. Chiral modes appear at boundaries where the superconductor is gapped by means of a magnetic insulator [17, 18]. The direct analogue of the leviton is the injection of single Majorana fermions into the edge modes [19–22].

An alternative route to flying qubits in a superconductor is to inject single *edge vortices* rather than single fermions [23]. Edge vortices are $\pi$-phase boundaries injected into the fermionic edge modes at a Josephson junction, in response to a $2\pi$ phase increment of the pair potential. (Recall that a fermionic phase shift is one-half the phase shift for Cooper pairs.) Unlike Majorana fermions, which are Abelian quasiparticles, the edge vortices are non-Abelian anyons: A qubit encoded in the fermion parity of a pair of edge vortices is a topologically protected degree of freedom, which can be transformed by braiding (exchange) and measured by fusion (merging) of the vortices.

Previous works studied the braiding of an edge vortex with a bulk vortex [23] and the non-Abelian fusion rule of edge vortices [24]. In these studies the dynamics of the edge vortices was ignored, by assuming that the time scale $L/v$ for the propagation through the system is small compared to the duration $t_{\rm inj}$ of the injection process. In the present work we relax that assumption, with a twofold objective: Firstly, to provide a time-resolved description of the charge transferred into a metal contact by the edge vortices. Secondly, to enable the braiding of the world lines of vortices on opposite edges. Taken together, these two objectives allow for the time-resolved electrical detection of chiral edge vortex braiding.

The outline of the paper is as follows. In the next Sec. 2 we briefly describe the effective edge Hamiltonian from Ref. [23], on which our analysis is based. The time dependent scattering theory is developed in Secs. 3–5, both in a fermionic and a bosonic formulation. We will work mainly in the fermionic description, but the bosonized scattering operator is helpful to make contact, in Sec. 6, with the conformal theory of edge vortices [25, 26]. We apply the scattering theory to the dynamics of the edge vortices in Secs. 7 and 8, where we analyse their fusion and braiding, aiming at the electrical detection. We conclude in Sec. 9.

## 2    Effective edge Hamiltonian

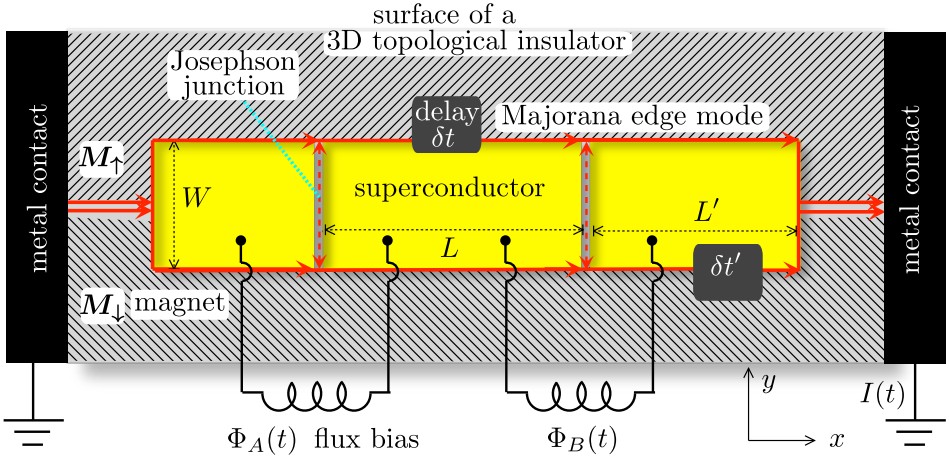

Figure 1: Geometry to create and braid two pairs of edge vortices in a topological insulator/magnetic insulator/superconductor heterostructure. The edge vortices are created at opposite ends of a Josephson junction, by an $h/2e$ flux bias $\Phi_{A,B}(t)$ that induces a $2\pi$ increment of the superconducting phase difference $\phi_{A,B}(t)$ across the junction. Each edge vortex contains a Majorana zero-mode and two zero-modes define a fermion-parity qubit. When two edge vortices are fused at the normal metal contact a current pulse $I(t)$ is produced, of integrated charge $Q = \pm e/2$. Gate electrodes on the edge modify the velocity of propagation and allow for a relative delay of vortices at upper and lower edge. This makes it possible to braid their world lines, as illustrated in Fig. 5. The braiding is a non-Abelian exchange operation which switches the fermion parity of the qubit and flips the sign of $Q$, allowing for electrical detection.

To set the stage, we summarize the findings of Ref. [23], with reference to the geometry of Fig. 1. A $2\pi$ increment of the phase shift $\phi(t) = (2e/\hbar)\Phi(t)$ across a flux-biased Josephson junction in a topological superconductor excites a vortex into each of the Majorana edge modes at opposite ends of the junction (at $y = \pm W/2$). The excitation process happens on the characteristic time scale

$$t_{\text{inj}} = (\xi_0/W)(d\phi/dt)^{-1}, \tag{2.1}$$

where $\xi_0 = \hbar v/\Delta_0$ is the superconducting coherence length (at Fermi velocity $v$ and gap

$\Delta_0$). We assume that $W/v \ll t_{\mathrm{inj}}$, so that the time for propagation along the junction (in the $y$-direction) can be neglected relative to the vortex injection time $t_{\mathrm{inj}}$. However, we will go beyond Ref. [23] to fully account for the finite propagation time along the edge (in the $x$-direction).

We will later introduce path length differences (or equivalently, velocity differences) between the upper and lower edge, but we first analyze the simplest case that the propagation time from one junction to the next is the same for both edge modes ($\delta t = \delta t' = 0$ in Fig. 1).

The effective Hamiltonian of the edge modes is given by [23]

$$H = iv \begin{pmatrix} -\partial/\partial x & -\delta(x)\alpha(t) \\ \delta(x)\alpha(t) & -\partial/\partial x \end{pmatrix} \equiv vp_x\sigma_0 + \delta(x)v\alpha(t)\sigma_y, \tag{2.2}$$

$$\alpha = \arccos\left(\frac{\cos(\phi/2) + \tanh\beta}{1 + \cos(\phi/2)\tanh\beta}\right) \times \mathrm{sign}\,(\phi), \quad \beta = \frac{W}{\xi_0}\cos(\phi/2). \tag{2.3}$$

(We have set $\hbar \equiv 1$.) The $2 \times 2$ Hermitian matrix $H$ acts on the Majorana fermion wave functions $\Psi = (\psi_1, \psi_2)$ at opposite edges of the superconductor, both propagating in the $+x$ direction. Since we take same velocity on both edges, the momentum operator $p_x = -i\partial/\partial x$ is multiplied by the unit matrix $\sigma_0$. The Josephson junction is positioned at $x = 0$ and couples the edges via the $\sigma_y$ Pauli matrix with a time dependent amplitude $\alpha(t)$. A $2\pi$ increment of $\phi$ corresponds to a $\pi$ increment of $\alpha$, in a step function manner when $W/\xi_0 \gg 1$,

$$\alpha(t) \approx \arccos[-\tanh(t/2t_{\mathrm{inj}})] \quad \text{if} \quad W \gg \xi_0. \tag{2.4}$$

Because the Hamiltonian $H$ is purely imaginary, the wave equation $\partial\psi/\partial t = -iH\psi$ is purely real — which is the defining property of a Majorana mode.

More generally, we can consider a sequence of Josephson junctions in series, at positions $x_1, x_2, \ldots$, each with its own phase difference $\phi_j(t)$ and corresponding $\alpha_j(t)$. We will also allow for bulk vortices in the superconductor. An $h/2e$ bulk vortex at $x = x_{\mathrm{vortex}}$ introduces a $\pi$ phase difference between the upper and lower edge modes "downstream" from the vortex (so for $x > x_{\mathrm{vortex}}$). This can be accounted for in $H$ by a term $(\pi/2)\delta(x - x_{\mathrm{vortex}})\sigma_z$, or equivalently, upon gauge transformation,[1] by switching the sign of the $\sigma_y$ term:

$$H = vp_x\sigma_0 + \sum_j (-1)^{n_j}\delta(x - x_j)v\alpha_j(t)\sigma_y. \tag{2.5}$$

Here $n_j$ is the number of vortices "upstream" from Josephson junction $j$ (so the number of vortices at $x < x_j$).

## 3    Construction of the phase field

The wave equation $i\partial\psi/\partial t = H\psi$ has the general solution

$$\psi(x, t) = e^{-i\Lambda(x,t)\sigma_y}\psi_0(x - vt), \tag{3.1}$$

---

[1]The gauge transformation $H \mapsto U^\dagger H U$ with $U = \exp[-i(\pi/2)\theta(x - x_{\mathrm{vortex}})\sigma_z]$ removes $(\pi/2)\delta(x - x_{\mathrm{vortex}})\sigma_z$ from $H$ and switches the sign of $\delta(x - x_j)\alpha(t)\sigma_y$ if $x_j > x_{\mathrm{vortex}}$. This gauge transformation ensures that the edge Hamiltonian remains purely imaginary in the presence of bulk vortices.

in terms of a phase field $\Lambda(x,t)$ determined by

$$(\partial_t + v\partial_x)\Lambda(x,t) = \sum_j (-1)^{n_j}\delta(x-x_j)v\alpha_j(t)$$

$$\Rightarrow \Lambda(x,t) = \sum_j (-1)^{n_j}\alpha_j(t - x/v + x_j/v)\theta(x-x_j). \tag{3.2}$$

(We abbreviate $\partial_q = \partial/\partial q$.) For an equivalent scalar solution, the two real components of $\psi = (\psi_1, \psi_2)$ (Majorana modes at upper and lower edge) can be combined into a complex wave function $\Psi = 2^{-1/2}(\psi_1 - i\psi_2)$ (a Dirac mode), which evolves in time as

$$\Psi(x,t) = e^{-i\Lambda(x,t)}\Psi_0(x - vt). \tag{3.3}$$

The phase field $\Lambda$ in the geometry of Fig. 1 is plotted in Fig. 2, as function of $x$ for a fixed $t$. A $2\pi$ increment of the phase of the pair potential $\Delta_0 e^{i\phi}$ creates a $\pi$-phase domain wall for Majorana fermions on the edge, propagating away from the Josephson junction with velocity $v$.

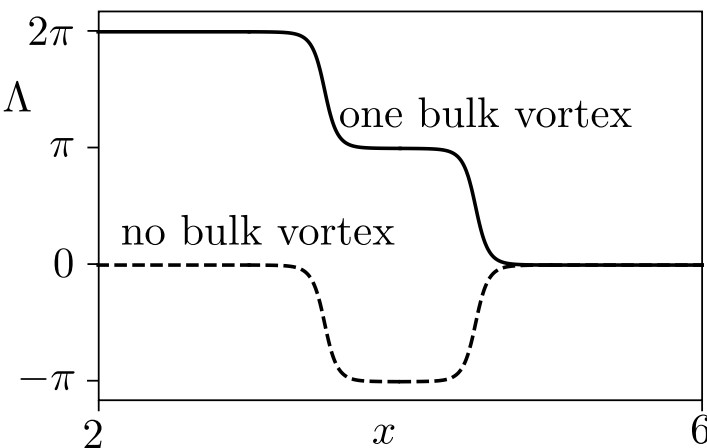

Figure 2: Phase field $\Lambda(x,t)$ of the Majorana edge modes, calculated from Eq. (3.2) for Josephson junctions at $x_1 = 0$ and $x_2 = 1$ and plotted as a function of $x$ for $t = 4$. The phase $\phi(t) = \phi_1(t) = -\phi_2(t)$ increases linearly from 0 at $t = 0$ to $2\pi$ at $t = 1$. The amplitude $\alpha(t)$ is calculated from Eq. (2.3) at $W/\xi_0 = 5$. Solid and dashed curves are with and without a bulk vortex in between the Josephson junctions. The $\pi$-phase domain walls propagate in the $+x$ direction with velocity $v = 1$.

The phase field determines the time dependent scattering matrix $S(t,t')$ that relates incoming and outgoing wave amplitudes. To formulate a scattering problem we assume that the Josephson junctions are all contained in the interval $0 < x < L$, so that the edge modes propagate freely for $x < 0$ [incoming state $\psi_{\text{in}}(t) = \psi(0,t)$] and for $x > L$ [outgoing state $\psi_{\text{out}}(t) = \psi(L,t)$]. The amplitudes are related by

$$\psi_{\text{out}}(t) = \int_{-\infty}^{\infty} S(t,t')\psi_{\text{in}}(t')\,dt', \quad S(t,t') = e^{-i\Lambda(t)\sigma_y}\delta(t' - t + L/v),$$

$$\Lambda(t) \equiv \Lambda(L,t) = \sum_j (-1)^{n_j}\alpha_j(t - L/v + x_j/v). \tag{3.4}$$

In the energy domain one has

$$S(\varepsilon, \varepsilon') = \int dt \int dt' \, e^{i\varepsilon t - i\varepsilon' t'} S(t, t') = e^{i\varepsilon' L/v} \int dt \, e^{i(\varepsilon - \varepsilon')t} e^{-i\Lambda(t)\sigma_y}. \tag{3.5}$$

The scattering matrix of Ref. [23] is recovered if the finite propagation time between the Josephson junctions is ignored.

Eqs. (3.4) and (3.5) relate the real Majorana fields $\psi_{\text{in}}$ and $\psi_{\text{out}}$. To relate the complex Dirac fields $\Psi_{\text{in}}$ and $\Psi_{\text{out}}$ one removes the $\sigma_y$ Pauli matrix that multiplies the phase field $\Lambda$.

## 4 Bosonized scattering operator

We proceed from the single-particle dynamics described by the scattering matrix (3.4) to the time evolution of the many-particle state in a bosonized formulation. This is not an essential step, all results can be obtained from the fermionic scattering matrix, but the bosonization provides for a direct route to the transferred charge in Sec. 5 and it will allow us to explicitly construct the vortex field operator in Sec. 6.

We transform to a coordinate frame that moves along the edge with velocity $v \equiv 1$, so the independent space and time variables are $s = x - t$ and $\tau = t + x$. In the complex representation $\Psi = 2^{-1/2}(\psi_1 - i\psi_2)$ of the edge modes the scalar wave equation reads $i\partial\Psi/\partial\tau = (\partial\Lambda/\partial\tau)\Psi$. The corresponding evolution of the many-particle state $|\tau\rangle$ is given in terms of the fermionic field operator $\hat{\Psi}(s)$ by

$$i\frac{\partial}{\partial\tau}|\tau\rangle = \hat{\mathcal{V}}(\tau)|\tau\rangle, \quad \hat{\mathcal{V}}(\tau) = \int ds \, \hat{\Psi}(s)^\dagger \hat{\Psi}(s)\partial_\tau\Lambda(s,\tau),$$
$$\Lambda(s,\tau) = \sum_j (-1)^{n_j}\alpha_j(x_j - s)\theta(s + \tau - 2x_j). \tag{4.1}$$

The scattering operator $\mathcal{S}$ that solves Eq. (4.1) for $|\tau\rangle = \hat{\mathcal{S}}(\tau)|0\rangle$ is given formally by

$$\hat{\mathcal{S}}(\tau) = \mathcal{T}\exp\left(-i\int_0^\tau d\tau'\hat{\mathcal{V}}(\tau')\right), \tag{4.2}$$

where $\mathcal{T}$ indicates time ordering of the exponential operator (later times to the left of earlier times). This expression still needs to be regularized, which is conveniently achieved by bosonization [4]. (See Ref. [27] for an alternative approach.)

The regularized density operator of the chiral fermionic mode is a Hermitian bosonic field $\hat{\rho}(s)$ defined by

$$\hat{\rho}(s) = : \hat{\Psi}^\dagger(s)\hat{\Psi}(s) : \tag{4.3}$$

The colons prescribe the subtraction of the (infinite) expectation value in the unperturbed Fermi sea. The anticommutator $\{\hat{\Psi}^\dagger(s), \hat{\Psi}(s')\} = \delta(s - s')$ of the fermionic field corresponds to the density commutator [28]

$$[\hat{\rho}(s), \hat{\rho}(s')] = \frac{i}{2\pi}\frac{\partial}{\partial s}\delta(s - s'). \tag{4.4}$$

The corresponding commutator of $\hat{\mathcal{V}}(\tau) = \int ds\, \rho(s)\partial_\tau\Lambda(s,\tau)$ is a c-number,

$$[\hat{\mathcal{V}}(\tau), \hat{\mathcal{V}}(\tau')] = -\frac{i}{2\pi}\int ds\left(\partial_s\partial_\tau\Lambda(s,\tau)\right)\partial_{\tau'}\Lambda(s,\tau'). \tag{4.5}$$

The Magnus expansion for a c-number commutator,

$$\mathcal{T}e^{-i\int_0^\tau d\tau'\,\hat{V}(\tau')} = e^{-i\int_0^\tau d\tau'\hat{V}(\tau')}e^{-\frac{1}{2}\int_0^\tau d\tau_1\int_0^{\tau_1} d\tau_2\,[\hat{V}(\tau_1),\hat{V}(\tau_2)]}, \tag{4.6}$$

allows us to remove the time ordering. The time integrals in the exponent of Eq. (4.2) can then be evaluated,

$$\mathcal{T}\exp\left(-i\int_0^\tau d\tau'\,\hat{V}(\tau')\right) = e^{i\varphi(\tau)}\exp\left(-i\int ds\,\hat{\rho}(s)\Lambda(s,\tau)\right),$$
$$\varphi(\tau) = \frac{1}{4\pi}\int ds\int_0^\tau d\tau'\Lambda(s,\tau')\partial_s\partial_{\tau'}\Lambda(s,\tau'). \tag{4.7}$$

One more step is needed. The operator $\hat{\rho}$ creates particle-hole excitations, preserving the fermion parity, so for a complete description of the scattering process we also need a Klein factor, an operator $\hat{F}$ that connects the ground states with $N$ and $N+1$ particles [28]:

$$\hat{F}|0\rangle_N = |0\rangle_{N+1}, \quad [\hat{\rho}(s),\hat{F}] = 0, \quad \hat{F}\hat{F}^\dagger = 1. \tag{4.8}$$

A fermion parity switch is possible because the edge vortices exchange a quasiparticle with each of the $N_{\text{vortex}}$ bulk vortices in between the Josephson junctions [23]. The final expression for the bosonized scattering operator is

$$\hat{\mathcal{S}}(\tau) = e^{i\varphi(\tau)}\hat{F}^{N_{\text{vortex}}}\exp\left(-i\int ds\,\hat{\rho}(s)\Lambda(s,\tau)\right). \tag{4.9}$$

## 5 Half-integer charge transfer

The operator $ev\hat{\rho}$ is the charge current density operator, regularized by subtracting the contribution from the unperturbed Fermi sea. Using the identity

$$\hat{\mathcal{S}}^\dagger(\tau)\hat{\rho}(s)\hat{\mathcal{S}}(\tau) = \hat{\rho}(s) + \frac{1}{2\pi}\frac{\partial}{\partial s}\Lambda(s,\tau), \tag{5.1}$$

which follows from Eqs. (4.4) and (4.9), we obtain the average current

$$I(s,\tau) = ev\langle\tau|\hat{\rho}(s)|\tau\rangle = ev\langle 0|\hat{\mathcal{S}}^\dagger(\tau)\hat{\rho}(s)\hat{\mathcal{S}}(\tau)|0\rangle = \frac{ev}{2\pi}\frac{\partial}{\partial s}\Lambda(s,\tau). \tag{5.2}$$

Assuming again that the Josephson junctions are in a finite interval $0 < x < L$, and considering the current of the outgoing state at $x > L$, we have $\Lambda(s,\tau) = \sum_j(-1)^{n_j}\alpha_j(x_j-s)$, hence (restoring the original variables $x, t$),

$$I(x,t) = -\frac{e}{2\pi}\sum_j(-1)^{n_j}\frac{\partial}{\partial t}\alpha_j(t - x/v + x_j/v). \tag{5.3}$$

Each $\pi$-phase domain wall carries a charge of $\pm e/2$. In total, the average transferred charge is 0 or $\pm e$ depending on whether $N_{\text{vortex}}$ is even or odd [23].

# 6 Construction of the vortex field operator

Given the phase field $\Lambda(x,t)$, we define the unitary operator

$$\hat{\mu}(x) = \exp\left(-i\int dx'\,\hat{\rho}(x')\Lambda(x',x/v)\right). \tag{6.1}$$

The commutator

$$[\hat{\rho}(x),\hat{\Psi}(x')] = [\hat{\Psi}^\dagger(x)\hat{\Psi}(x),\hat{\Psi}(x')] = -\delta(x-x')\hat{\Psi}(x) \tag{6.2}$$

implies that[2]

$$\hat{\mu}(x)\hat{\Psi}(x') = e^{i\Lambda(x',x/v)}\hat{\Psi}(x')\hat{\mu}(x). \tag{6.3}$$

To interpret this relation we consider the regime $W \gg \xi_0$ when each $\pi$-phase boundary in Fig. 2 becomes a step function. For a single Josephson junction at $x = 0$ and a phase difference $\phi(t)$ which crosses $\pi$ at $t = 0$ the phase field is

$$\Lambda(x',t) = \pi\theta(vt - x')\theta(x'). \tag{6.4}$$

Eq. (6.3) takes the form

$$\hat{\mu}(x)\hat{\Psi}(x') = \begin{cases} -\hat{\Psi}(x')\hat{\mu}(x) & \text{if } 0 < x' < x, \\ +\hat{\Psi}(x')\hat{\mu}(x) & \text{otherwise.} \end{cases} \tag{6.5}$$

In the basis of Majorana fermion fields $\hat{\psi}_1(x)$, $\hat{\psi}_2(x)$ on upper and lower edge, with anti-commutator $\{\hat{\psi}_n(x),\hat{\psi}_m(x')\} = \delta_{nm}\delta(x-x')$, the vortex field operator (6.1) may be written as

$$\hat{\mu}(x) = \exp\left(-\int dx'\,\hat{\psi}_1(x')\hat{\psi}_2(x')\Lambda(x',x/v)\right). \tag{6.6}$$

The commutator (6.5) applies to each Majorana fermion field separately,

$$\hat{\mu}(x)\hat{\psi}_n(x') = \begin{cases} -\hat{\psi}_n(x')\hat{\mu}(x) & \text{if } 0 < x' < x, \\ +\hat{\psi}_n(x')\hat{\mu}(x) & \text{otherwise.} \end{cases} \tag{6.7}$$

The commutator (6.7) is the defining property of a vortex field operator, such as the twist field in the conformal field theory[3] of Majorana edge modes [25, 26]. The step function approximation (6.4) of the phase field $\Lambda$ corresponds to the neglect of the finite size of the core of the edge vortex. In that zero-core limit $\hat{\mu}(x)$ is both unitary and Hermitian (it squares to the identity). More generally, the vortex field operator (6.1) is unitary but not Hermitian.

# 7 Fusion of edge vortices with a relative time delay

So far we have assumed that the vortices propagate along opposite edges with the same velocity. We now relax that assumption and allow for a relative time delay between upper and lower edge. (This delay is crucial for the braiding scheme of Fig. 1, which we will study

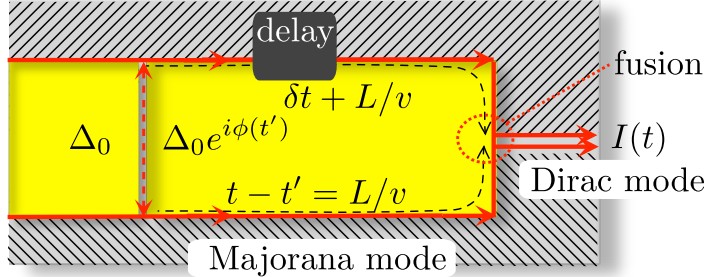

Figure 3: Geometry described by the scattering matrix (7.1).

in Sec. 8.) Here we present a calculation using the scattering matrix, an alternative Green's function calculation is given in App. A.

To study the effect of a time delay on the fusion of two edge vortices it is sufficient to consider a single Josephson junction, as in Fig. 3. The junction is at $x = 0$, with phase difference $\phi(t)$ and corresponding scattering phase $\alpha(t)$. The propagation time from $x = 0$ to $x = L$ along the upper and lower edge is $L/v + \delta t$ and $L/v$, respectively, corresponding to the scattering matrix

$$S(t, t') = \begin{pmatrix} \delta(t' - t + L/v + \delta t) & 0 \\ 0 & \delta(t' - t + L/v) \end{pmatrix} e^{-i\alpha(t')\sigma_y}. \tag{7.1}$$

For $\delta t = 0$ this reduces to the previous Eq. (3.4). At $x = L$ the two Majorana modes merge to form a single Dirac mode, which carries an electrical current into a normal metal contact.

The expectation value $I(t)$ of the current can be calculated starting from a scattering formula in the energy domain,

$$I(t) = e \int_{-\infty}^{\infty} \frac{dE}{2\pi} \int_{-\infty}^{\infty} \frac{dE'}{2\pi} \int_{-\infty}^{\infty} \frac{d\omega}{2\pi} e^{i\omega t} f(E')[1 - f(E)]$$
$$\times \operatorname{Tr} S^\dagger(E + \omega/2, E')\sigma_y S(E - \omega/2, E'), \tag{7.2}$$

which says that the current is produced by scattering from filled states with weight $f(E')$ to empty states with weight $1 - f(E)$. (The function $f(E) = (1 + e^{E/k_\mathrm{B}T})^{-1}$ is the equilibrium Fermi function at temperature $T$.) In App. B we derive the equivalent time-domain expression at zero temperature,[4]

$$I(t) = \frac{ie}{4\pi} \int_{-\infty}^{\infty} dt' \int_{-\infty}^{\infty} dt'' \frac{1}{t'' - t'} \operatorname{Tr} S^\dagger(t, t')\sigma_y S(t, t''). \tag{7.3}$$

Substitution of Eq. (7.1) into Eq. (7.3) gives the result

$$I(t) = \frac{e}{2\pi} \frac{\sin[\alpha(t - L/v - \delta t) - \alpha(t - L/v)]}{\delta t}. \tag{7.4}$$

---

[2]If we define $\hat{O}(\xi) = e^{i\xi \int dx' \, \hat{\rho}(x')\Lambda(x',t)} \hat{\Psi}(x) e^{-i\xi \int dx' \, \hat{\rho}(x')\Lambda(x',t)}$ then $\partial_\xi \hat{O}(\xi) = -i\Lambda(x,t)\hat{O}(\xi)$, hence $\hat{O}(\xi) = e^{-i\xi\Lambda(x,t)}\hat{O}(0)$. The result (6.3) then follows at $\xi = 1$.

[3]In the context of the 2D Ising model the commutator (6.7) defines the socalled disorder field [29].

[4]App. B also shows how to regularize the singularity at $t'' - t'$ in Eq. (7.3). For our applications no regularization is needed.

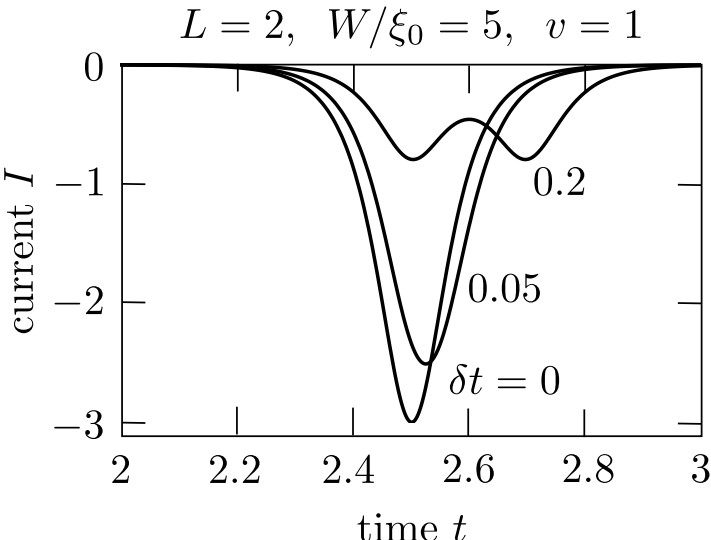

Figure 4: Time dependent current $I(t)$ (in dimensionless units) produced by the fusion of two edge vortices in the geometry of Fig. 3. The phase $\phi(t)$ across the single Josephson junction increases linearly from 0 at $t = 0$ to $2\pi$ at $t = 1$. The amplitude $\alpha(t)$ is calculated from Eq. (2.3) at $W/\xi_0 = 5$, so that $t_{\text{inj}} = (10\pi)^{-1} \approx 0.03$. The three curves, calculated from Eq. (7.4), correspond to different values of the relative delay $\delta t$ between edge vortices on upper and lower edge. The current pulse is suppressed when $\delta t \gg t_{\text{inj}}$.

plotted in Fig. 4. When the relative delay time vanishes we recover the expected limit

$$\lim_{\delta t \to 0} I(t) = -\frac{e}{2\pi} \frac{d}{dt} \alpha(t - L/v), \tag{7.5}$$

in accord with Eq. (5.3) for a single Josephson junction without bulk vortices.

The average transferred charge $Q = \int I(t)dt$ decays from $e/2$ to zero when $\delta t$ becomes large compared to the injection time $t_{\text{inj}}$. If we take the large $W/\xi_0$ functional form (2.4) for $\alpha(t)$ we have a simple analytical expression,

$$Q(t) = -\frac{e}{2} \frac{\tanh(\delta t/4t_{\text{inj}})}{\delta t/4t_{\text{inj}}}. \tag{7.6}$$

# 8 Braiding of edge vortices with a relative time delay

Now that we have a time-resolved scattering theory of edge vortices we can describe the braiding of their world lines. We will consider separately the braiding of vortices propagating in the same direction or in the opposite direction.

## 8.1 Co-propagating edge vortices

The world lines of vortices moving in the same direction on opposite edges can be braided by introducing a delay, as indicated in the geometry of Fig. 1. The braiding diagram is shown

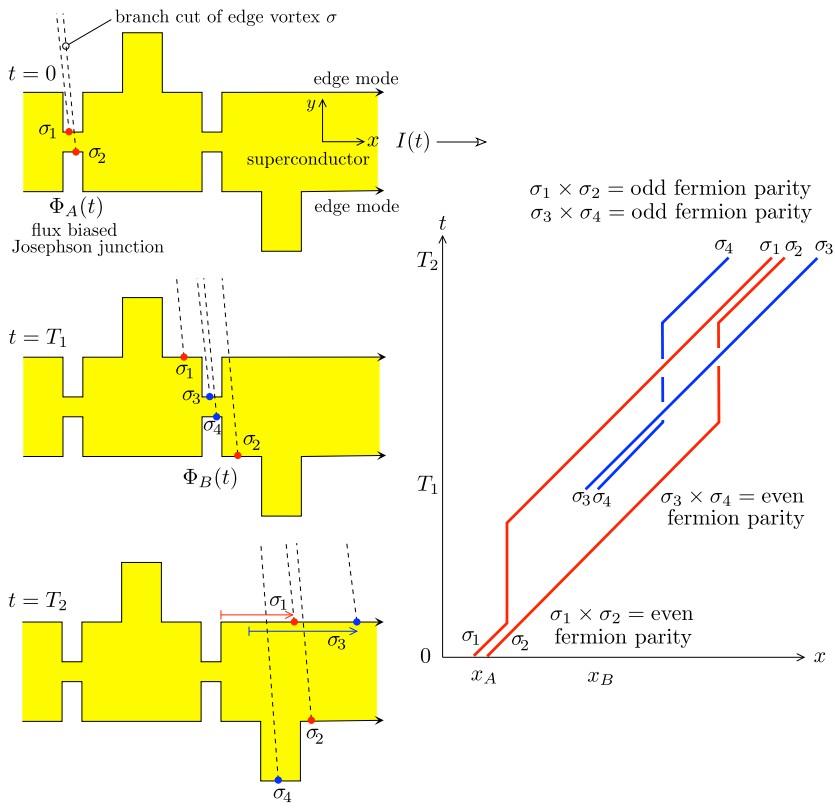

Figure 5: Braiding of vortices moving in the same direction. The three diagrams at the left show the edge vortices at three instants in time; the vortices are produced by an $h/2e$ flux increment, first at Josephson junction $A$ (time $t = 0$) and then at Josephson junction $B$ (time $t = T_1$). Each vortex induces a $2\pi$ phase shift of the order parameter across a branch cut, indicated by the dashed lines. The Majorana operator $\gamma_n$ associated with vortex $\sigma_n$ changes sign when the vortex crosses a branch cut from some other vortex. This happens once for vortex 1 and twice for vortex 3, so $\gamma_1$ changes sign but $\gamma_3$ does not. The vortices 2 and 4 do not cross a branch cut, so $\gamma_2$ and $\gamma_4$ are unaffected. In the space-time braiding diagram the crossing of a branch cut is indicated by an overpass. At the end of this process both fermion parity operators $i\gamma_1\gamma_2$ and $i\gamma_3\gamma_4$ change sign. Hence two fermions are produced, one shared by vortices 1 and 2 and one shared by vortices 3 and 4. This can be detected electrically as a sign change of the current pulse $I(t)$ produced by the fusion of vortices 1 and 2 when they enter a metal contact.

in Fig. 5, where the delay is indicated schematically as a path length difference (a velocity difference would have an equivalent effect).

We extend the calculation of Sec. 7 to include two Josephson junctions (scattering phases $\alpha_A$ and $\alpha_B$), and two delay times: $\delta t$ on the upper edge between the first and second junction, and $\delta t'$ on the lower edge after the second junction. The braiding exchanges a fermion between vortex pair 1,2 produced at the first Josephson junction and vortex pair 3,4 from the second Josephson junction, switching the fermion parity of the two vortex pairs from even–even to odd–odd. As we will now show, the fermion parity switch can be detected electrically as a switch in the sign of the current peak, from integrated charge $-e/2$ to $+e/2$.

The scattering matrix corresponding to the geometry of Fig. 1 is

$$S(t,t') = \int_{-\infty}^{\infty} dt'' \begin{pmatrix} \delta(t'' - t + L'/v) & 0 \\ 0 & \delta(t'' - t + L'/v + \delta t') \end{pmatrix} e^{-i\alpha_B(t'')\sigma_y}$$
$$\cdot \begin{pmatrix} \delta(t' - t'' + L/v + \delta t) & 0 \\ 0 & \delta(t' - t'' + L/v) \end{pmatrix} e^{-i\alpha_A(t')\sigma_y}. \tag{8.1}$$

Substitution into Eq. (7.3) gives, in the limit $\delta t' \to \delta t$, the time dependent current

$$I(t) = -\frac{e}{2\pi} \cos\alpha_B(t_B)\cos\alpha_B(t_B - \delta t)\frac{d\alpha_A(t_A - \delta t)}{dt_A}$$
$$- \frac{e}{2\pi\delta t}\Bigg\{ \sin\alpha_B(t_B)\cos\alpha_B(t_B - \delta t)\cos[\alpha_A(t_A) - \alpha_A(t_A - \delta t)]$$
$$- \sin\alpha_B(t_B - \delta t)\cos\alpha_B(t_B)\cos[\alpha_A(t_A - \delta t) - \alpha_A(t_A - 2\delta t)]$$
$$+ \tfrac{1}{2}\sin\alpha_B(t_B - \delta t)\sin\alpha_B(t_B)\sin[\alpha_A(t_A) - \alpha_A(t_A - 2\delta t)]\Bigg\}, \tag{8.2}$$

with $t_A = t - L/v - L'/v$, $t_B = t - L'/v$. As a check, we can send $\delta t \to 0$ and recover the expected $I(t) = -(e/2\pi)[\alpha'_A(t_A) + \alpha'_B(t_B)]$.

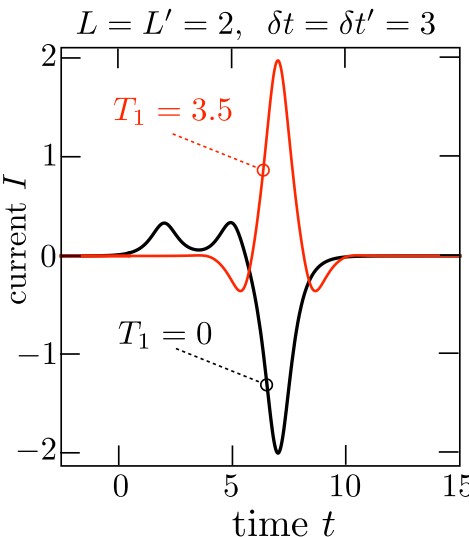

Figure 6: Time dependent current $I(t)$ (in dimensionless units) in the geometry of Fig. 1. The superconducting phase is incremented from 0 to $2\pi$ across Josephson junction $A$ at time $t = 0$, and then back from $2\pi$ to 0 across Josephson junction $B$ at time $t = T_1$. The curves are calculated from Eq. (8.2), with $\alpha_A(t) = \arccos(-\tanh 2t)$ and $\alpha_B(t) = -\alpha_A(t - T_1)$. For the black curve we took $T_1 = 0$, while for the red curve we introduced a delay $T_1 = 3.5$. The resulting sign switch of the current pulse signals the braiding of the world lines of the injected vortices, as indicated in Fig. 5. The large peak (integrated charge $\pm e/2$) is from the fusion of vortices 1 and 2, the small side peaks come from vortices 3 and 4, which have little overlap and therefore only give a small contribution to the transferred charge.

With reference to Figs. 1 and 5, the vortices at junction $A$ are injected at time $t = 0$ and those at junction $B$ are injected at a later time $t = T_1$ such that $L/v < T_1 < L/v + \delta t$. At

the time $t = T_2 = L/v + L'/v + \delta t$ one thus has $t_B > T_1$ and $t_B - \delta t < T_1$, hence $\alpha_B(t_B) \approx 0$ and $\alpha_B(t_B - \delta t) \approx \pi$. Inspection of Eq. (8.2) shows that the term between curly brackets is suppressed, leaving only the first term with a switched sign:

$$I(T_2) \approx -\frac{e}{2\pi}\cos\alpha_B(t_B)\cos\alpha_B(t_B - \delta t)\alpha'_A(t_A - \delta t) \approx +\frac{e}{2\pi}\alpha'_A(0). \tag{8.3}$$

In Fig. 6 we show how the sign switch follows from the full Eq. (8.2).

## 8.2   Counter-propagating edge vortices

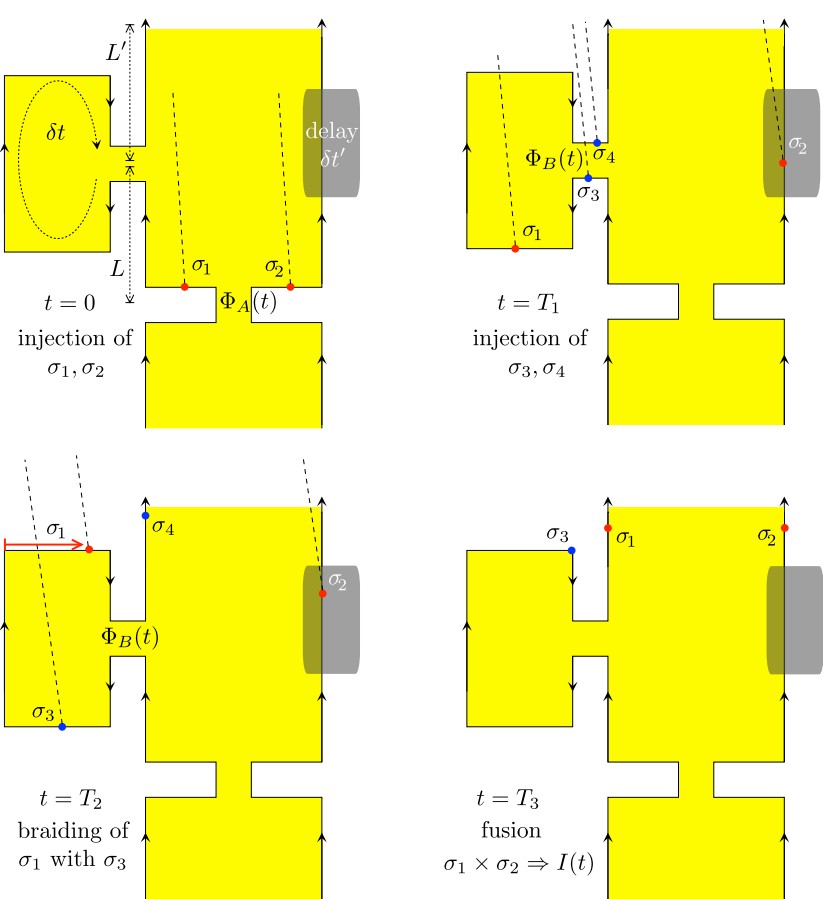

Figure 7: Four steps in the braiding of vortices $\sigma_1$ and $\sigma_3$ moving in opposite directions.

An alternative diagram to braid vortices moving in opposite directions is shown in Fig. 7. The first Josephson junction $A$ is the same as before, with scattering matrix $S_A = e^{-i\alpha_A\sigma_y}$ depending on a parameter $\alpha_A$ given by Eq. (2.3). A $2\pi$ increment of the phase difference $\phi_A$ across junction $A$ injects edge vortices $\sigma_1$ and $\sigma_2$.

The second Josephson junction $B$ injects vortices $\sigma_3$ and $\sigma_4$ in response to a $2\pi$ increment of $\phi_B$. Its scattering matrix $S_B$ has a different form than $S_A$, because junction $B$ couples counter-propagating edge modes while junction $A$ couples co-propagating modes. As discussed in Ref. [23], the difference manifests itself in the symmetry relation $S_A(\phi_A) = -S_A(\phi_A + 2\pi)$

versus $S_B(\phi_B) = -S_B^{\mathrm{T}}(\phi_B + 2\pi)$. The corresponding expression for $S_B$ is [17]

$$S_B = \begin{pmatrix} \tanh \beta_B & 1/\cosh \beta_B \\ -1/\cosh \beta_B & \tanh \beta_B \end{pmatrix}, \quad \beta_B = \frac{W}{\xi_0} \cos(\phi_B/2). \tag{8.4}$$

The scattering matrix $S$ of the entire system is composed from $S_A$ and $S_B$, upon accounting for the time delays due to propagation along the edge. This gives an expression of the form

$$S(t, t') = \begin{pmatrix} \delta(t_A - t' - \delta t') & 0 \\ 0 & \sum_{n=0}^{\infty} S_n(t_B) \delta(t_A - t' - n\delta t) \end{pmatrix} e^{-i\alpha_A(t')\sigma_y}, \tag{8.5}$$

with the definitions $t_A = t - L/v - L'/v$, $t_B = t - L'/v$, and

$$S_0(t) = -\frac{1}{\cosh \beta_B(t)}, \quad S_1(t) = \tanh \beta_B(t) \tanh \beta_B(t - \delta t),$$

$$S_n(t) = \tanh \beta_B(t) \tanh \beta_B(t - n\delta t) \prod_{p=1}^{n-1} \frac{1}{\cosh \beta_B(t - p\delta t)}, \quad n \geq 2. \tag{8.6}$$

The delay $\delta t$ is the time it takes to circulate from junction $B$ back to the same junction (as indicated in the top left panel of Fig. 7). The sum over $n$ counts the number of times a vortex circulates around this delay loop. The delay $\delta t'$ at the opposite edge is adjustable by variation of the edge velocity.

Substitution into Eq. (7.3) gives the current

$$I(t) = -\frac{e}{2\pi} \sum_{n=0}^{\infty} \frac{S_n(t_B)}{n\delta t - \delta t'} \sin[\alpha_A(t_A - \delta t') - \alpha_A(t_A - n\delta t)]. \tag{8.7}$$

Note that $I(t) \equiv 0$ when $\alpha_A \equiv 0$, so when there is no vortex injection at junction $A$. In contrast to the case considered in Sec. 8.1, the vortices $\sigma_3$ and $\sigma_4$ injected at junction $B$ cannot transfer any charge into the metal contact, because they represent phase boundaries in a *single* Majorana edge mode. A minimum of two Majorana modes is needed for a nonzero charge transfer.

For non-overlapping vortices, when $t_{\mathrm{inj}} \ll \delta t$, the sum over $n$ in Eq. (8.7) converges rapidly, with the $n = 1$ term giving the dominant contribution. In the limit $\delta t \to \delta t'$ this results in the current

$$I(t) \approx -\frac{e}{2\pi} \tanh \beta_B(t_B) \tanh \beta_B(t_B - \delta t) \frac{d}{dt_A} \alpha_A(t_A - \delta t). \tag{8.8}$$

The vortices $\sigma_3, \sigma_4$ at junction $B$ are injected at time $t = T_1$ with $L/v < T_1 < L/v + \delta t$, when vortex $\sigma_1$ is inside the delay loop. At the fusion time $t = T_3 = L/v + L'/v + \delta t$ one thus has $t_B > T_1$ and $t_B - \delta t < T_1$, hence $\tanh \beta_B(t_B) \tanh \beta_B(t_B - \delta t) \approx -1$ — while if no vortices are injected when $\sigma_1$ is inside the delay loop one has $\tanh \beta_B(t_B) \tanh \beta_B(t_B - \delta t) \approx +1$. In Fig. 8 we show how the sign switch follows from Eq. (8.7).

# 9 Conclusion

In summary, we have shown how the braiding of world lines of edge vortices can be detected in electrical conduction. The signature of the non-Abelian exchange is the transfer of a fermion

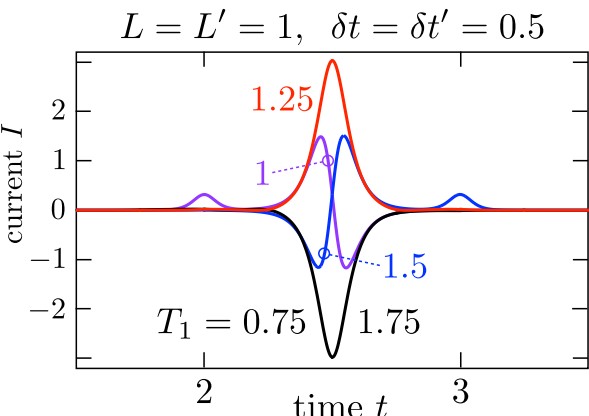

Figure 8: Same as Fig. 6, but now for the geometry of Fig. 7. The curves are calculated from Eq. (8.7), with $\alpha_A(t)$ given by Eq. (2.3) and $\beta_B(t)$ given by Eq. (8.4). We took $W/\xi_0 = 5$ and incremented $\phi_A, \phi_B$ by $2\pi$ with a constant rate $d\phi/dt = 2\pi$. The curves are for five different values of $T_1$ (the curves for $T_1 = 0.75$ and 1.75 are indistinguishable). The $\pm e/2$ current pulse from the fusion of vortices $\sigma_1$ and $\sigma_2$ changes sign when $T_1$ is in the interval $(L, L + \delta t) = (1, 1.5)$ in which $\sigma_1$ is braided with $\sigma_3$.

from one vortex pair to another, which is detected as a sign change of the current pulse when two vortices are fused in a metal contact.

The edge vortices are elementary excitations of a chiral Majorana edge mode in a topological superconductor, and it is instructive to make a comparison with the elementary excitations of the chiral Dirac edge modes in a quantum Hall insulator [9]. In that context the leviton is the charge-$e$ excitation of minimal noise, produced by a $2\pi$ phase increment of the single-electron wave function [4–6]. The edge vortices, in contrast, are injected by a $2\pi$ phase increment of the pair potential, which is a $\pi$ phase shift for single fermions. This explains why the elementary current pulse transfers *half-integer* charge.

In a different context, the fractionally charged $\pi$-phase domain wall bound to the edge vortices is the mobile counterpart of the $\pm e/2$ charge bound to a zero-mode in a topological insulator [30, 31]. For example, in a narrow ribbon of quantum spin Hall insulator a $\pm e/2$ domain wall is formed by the merging of $\pm e/4$ charges on opposite edges of the ribbon [32]. In a topological superconductor the $\pm e/4$ charge associated with a vortex is referred to as its "topological spin" [33, 34].

Since only integer charge can enter into a normal metal, the fractional charge transfer by edge vortices cannot be noiseless — that is a basic distinction with single-electron levitons. For applications to quantum information processing, it is relevant that the charge noise only appears when the edge vortices are fused. The qubit degree of freedom, the fermion parity, is topologically protected as long as the vortices remain widely separated.

## Acknowledgements

A. R. Akhmerov suggested to us the vortex braiding geometry of Fig. 5. This research was supported by the Netherlands Organization for Scientific Research (NWO/OCW) and by the European Research Council (ERC).

# A  Current expectation value from Green's function of a chiral mode

The Green's function of a chiral mode $\Psi_0(x)$ of free fermions is

$$\langle \Psi_0^\dagger(x+d)\Psi_0(x)\rangle = \langle \Psi_0(x+d)\Psi_0^\dagger(x)\rangle = \int_0^\infty \frac{dk}{2\pi}e^{ikd} = \frac{i}{2\pi}\frac{1}{d+i0^+}, \tag{A.1}$$

where $\langle\cdots\rangle$ is the equilibrium expectation value at zero temperature. We can use this Green's function for an alternative derivation of the time dependent current (7.4).

A relative delay $\tau$ in propagation time between upper and lower edge is introduced by the operator $\mathcal{D}(\tau) = e^{-(\tau/2)\sigma_z\partial_t}$ in the Majorana basis $\{\psi_1,\psi_2\}$, corresponding to $\mathcal{D}(\tau) = e^{-(\tau/2)\nu_x\partial_t}$ in the electron-hole basis $\{\Psi,\Psi^\dagger\}$. (We use different symbols $\sigma$ and $\nu$ to distinguish Pauli matrices in the two bases.) The chiral mode evolves in the single-junction geometry of Fig. 3 as

$$\begin{pmatrix}\Psi(x,t)\\\Psi^\dagger(x,t)\end{pmatrix} = \mathcal{D}(\tau)e^{-i\alpha(t-x/v)\nu_z}\begin{pmatrix}\Psi_0(x-vt)\\\Psi_0^\dagger(x-vt)\end{pmatrix}. \tag{A.2}$$

In view of the identity

$$U\nu_z U^\dagger = \nu_x, \quad U = 2^{-1/2}\begin{pmatrix}1 & -i\\1 & i\end{pmatrix}, \tag{A.3}$$

we have

$$\begin{aligned}
\begin{pmatrix}\Psi(x,t)\\\Psi^\dagger(x,t)\end{pmatrix} &= Ue^{-(\tau/2)\nu_z\partial_t}U^\dagger e^{-i\alpha(t-x/v)\nu_z}\begin{pmatrix}\Psi_0(x-vt)\\\Psi_0^\dagger(x-vt)\end{pmatrix}\\
\Rightarrow \Psi(x,t) &= -\tfrac{1}{2}e^{i\alpha(t+\tau/2-x/v)}\Psi_0^\dagger(x+v\tau/2) + \tfrac{1}{2}e^{i\alpha(t-\tau/2-x/v)}\Psi_0^\dagger(x-v\tau/2)\\
&\quad + \tfrac{1}{2}e^{-i\alpha(t+\tau/2-x/v)}\Psi_0(x+v\tau/2) + \tfrac{1}{2}e^{-i\alpha(t-\tau/2-x/v)}\Psi_0(x-v\tau/2)\\
\Rightarrow \Psi^\dagger(x,t)\Psi(x,t) &= \tfrac{1}{2}e^{i\alpha(t+\tau/2-x/v)-i\alpha(t-\tau/2-x/v)}\Psi_0^\dagger(x+v\tau/2)\Psi_0(x-v\tau/2)\\
&\quad - \tfrac{1}{2}e^{-i\alpha(t+\tau/2-x/v)+i\alpha(t-\tau/2-x/v)}\Psi_0(x+v\tau/2)\Psi_0^\dagger(x-v\tau/2)\\
&\quad + \mathcal{O}(\Psi_0^\dagger\Psi_0^\dagger) + \mathcal{O}(\Psi_0\Psi_0).
\end{aligned} \tag{A.4}$$

The bilinears $\Psi_0^\dagger\Psi_0^\dagger$ and $\Psi_0\Psi_0$ vanish upon taking the expectation value. What remains is

$$\langle \Psi^\dagger(x,t)\Psi(x,t)\rangle = \frac{1}{2\pi}\sin\left[\alpha(t-\tau/2-x/v)-\alpha(t+\tau/2-x/v)\right]\frac{1}{v\tau+i0^+}. \tag{A.5}$$

Eq. (7.4) (with a relative delay $\delta t = \tau$) then follows from $I(t) = ev\langle\Psi^\dagger(L,t)\Psi(L,t)\rangle$.

# B  Derivation of the scattering formula (7.3) for the average current

The expectation value of the time-dependent electrical current is given in terms of the energy dependent scattering matrix by

$$I(t) = \frac{1}{2}e\int_{-\infty}^\infty \frac{dE}{2\pi}\int_{-\infty}^\infty \frac{dE'}{2\pi}\int_{-\infty}^\infty \frac{d\omega}{2\pi} f(E')e^{i\omega t}\,\mathrm{Tr}\,S^\dagger(E+\omega/2,E')\sigma_y S(E-\omega/2,E'). \tag{B.1}$$

The double counting of electrons and holes is corrected by the $1/2$ prefactor.

Because of unitarity, the integral (B.1) over $E'$ without the Fermi function $f(E')$ is proportional to $\delta(\omega) \operatorname{Tr} \sigma_y = 0$, so we may rewrite the expression identically as

$$
\begin{aligned}
I(t) = \frac{e}{2} \int_{-\infty}^{\infty} \frac{dE}{2\pi} \int_{-\infty}^{\infty} \frac{dE'}{2\pi} \int_{-\infty}^{\infty} \frac{d\omega}{2\pi}\, e^{i\omega t}[f(E') - f(E)] \\
\times \operatorname{Tr} S^\dagger(E + \omega/2, E')\sigma_y S(E - \omega/2, E') \\
= \frac{e}{2} \int_{-\infty}^{\infty} \frac{dE}{2\pi} \int_{-\infty}^{\infty} \frac{dE'}{2\pi} \int_{-\infty}^{\infty} \frac{d\omega}{2\pi}\, e^{i\omega t}[f(E')f(-E) - f(-E')f(E)] \\
\times \operatorname{Tr} S^\dagger(E + \omega/2, E')\sigma_y S(E - \omega/2, E'),
\end{aligned}
\tag{B.2}
$$

where in the second equality we used that $f(-E) = 1 - f(E)$.

Particle-hole symmetry in the Majorana basis, $S(E, E') = S^*(-E, -E')$, implies that the trace in Eq. (B.2) changes sign if $E, E' \mapsto -E, -E'$:

$$
\operatorname{Tr} S^\dagger(E + \omega/2, E')\sigma_y S(E - \omega/2, E') = -\operatorname{Tr} S^\dagger(-E + \omega/2, -E')\sigma_y S(-E - \omega/2, -E').
\tag{B.3}
$$

Hence the two terms in Eq. (B.2) combine into a single term, canceling the $1/2$,

$$
I(t) = e \int_{-\infty}^{\infty} \frac{dE}{2\pi} \int_{-\infty}^{\infty} \frac{dE'}{2\pi} \int_{-\infty}^{\infty} \frac{d\omega}{2\pi}\, e^{i\omega t} f(E')f(-E) \operatorname{Tr} S^\dagger(E + \omega/2, E')\sigma_y S(E - \omega/2, E').
\tag{B.4}
$$

This equation says that the current is produced by scattering from filled states with weight $f(E')$ to empty states with weight $f(-E) = 1 - f(E)$, as expected.

The Fourier transform from the energy to the time domain is defined by

$$
S(t', t) = \int_{-\infty}^{\infty} \frac{dE'}{2\pi} \int_{-\infty}^{\infty} \frac{dE}{2\pi}\, e^{-iE't'} S(E', E) e^{iEt}, \quad f(t) = \int_{-\infty}^{\infty} \frac{dE}{2\pi}\, e^{-iEt} f(E),
\tag{B.5}
$$

resulting in

$$
I(t) = e \iiint_{-\infty}^{\infty} dt_1 dt_2 dt_3\, f(t_1)f(t_2) \operatorname{Tr} S^\dagger(t - t_2/2, t_3 - t_1/2)\sigma_y S(t + t_2/2, t_3 + t_1/2).
\tag{B.6}
$$

We now take the zero-temperature limit. At $T = 0$ the Fermi function $f(E) = \theta(-E)$ has Fourier transform

$$
f(t) = \int_{-\infty}^{0} \frac{dE}{2\pi}\, e^{-iEt} = \frac{1}{2}\delta(t) + \frac{i}{2\pi t},
\tag{B.7}
$$

where the second term is a principal value. Because $S(t, t')$ is real in the Majorana basis, and $\sigma_y$ is imaginary, only the imaginary part of $f(t_1)f(t_2)$ contributes to the current, which equals

$$
\operatorname{Im} f(t_1)f(t_2) = \frac{1}{4\pi} \left( t_2^{-1}\delta(t_1) + t_1^{-1}\delta(t_2) \right).
\tag{B.8}
$$

Substitution into Eq. (B.6) gives $I(t)$ as the difference of two terms,

$$
\begin{aligned}
I(t) = \frac{ie}{4\pi} \int_{-\infty}^{\infty} dt' \int_{-\infty}^{\infty} \frac{d\tau}{\tau} \Big[ &\operatorname{Tr} S^\dagger(t, t' - \tau/2)\sigma_y S(t, t' + \tau/2) \\
&- \operatorname{Tr} S^\dagger(t + \tau/2, t')\sigma_y S(t - \tau/2, t') \Big].
\end{aligned}
\tag{B.9}
$$

Because of unitarity, the integral over $t'$ in the second term vanishes, leaving the first term, which is Eq. (7.3) in the main text.

For some applications it is helpful to retain the second term Eq. (B.9), since that regularizes the integrand at $\tau = 0$. In particular, we need both terms if we take the instantaneous scattering (adiabatic) limit *before* carrying out the time integration, replacing $S(t, t') \mapsto S_{\mathrm{F}}(t)\delta(t - t')$ with $S_{\mathrm{F}}(t)$ the "frozen" scattering matrix. In this limit

$$
\begin{aligned}
&\frac{1}{\tau} \operatorname{Tr} \left[ S^\dagger(t, t' - \tau/2)\sigma_y S(t, t' + \tau/2) - S^\dagger(t + \tau/2, t')\sigma_y S(t - \tau/2, t') \right] \\
&\mapsto \frac{1}{\tau}\delta(t - t' + \tau/2)\delta(t - t' - \tau/2) \operatorname{Tr} \left[ S_{\mathrm{F}}^\dagger(t)\sigma_y S_{\mathrm{F}}(t) - S_{\mathrm{F}}^\dagger(t + \tau/2)\sigma_y S_{\mathrm{F}}(t - \tau/2) \right] \\
&= \frac{1}{2}\delta(t - t')\delta(\tau) \operatorname{Tr} \left[ S_{\mathrm{F}}^\dagger(t)\sigma_y \frac{dS_{\mathrm{F}}(t)}{dt} - \frac{dS_{\mathrm{F}}^\dagger(t)}{dt}\sigma_y S_{\mathrm{F}}(t) \right] \\
&= \delta(t - t')\delta(\tau) \operatorname{Tr} S_{\mathrm{F}}^\dagger(t)\sigma_y \frac{dS_{\mathrm{F}}(t)}{dt}.
\end{aligned} \tag{B.10}
$$

The last equality follows from unitarity of $S_{\mathrm{F}}(t)$. Substitution into Eq. (B.9) then recovers the Brouwer formula [35],

$$
I(t) = \frac{ie}{4\pi} \operatorname{Tr} S_{\mathrm{F}}^\dagger(t)\sigma_y \frac{\partial}{\partial t} S_{\mathrm{F}}(t). \tag{B.11}
$$

Eq. (7.3) can be seen as a generalization of the Brouwer formula beyond the adiabatic regime.

Two further remarks about this scattering formula:

- We have assumed chiral conduction, but we may generalize to a situation with backscattering by inserting a projector $P_{\mathrm{out}}$ into the outgoing lead,

$$
I(t) = \frac{ie}{4\pi} \int_{-\infty}^{\infty} dt' \int_{-\infty}^{\infty} dt'' \frac{1}{t'' - t'} \operatorname{Tr} S^\dagger(t, t') P_{\mathrm{out}} \sigma_y S(t, t''). \tag{B.12}
$$

- In applications without superconductivity, it is more natural to work in the electron-hole basis, where $\sigma_y$ is transformed into $\sigma_z$. When the scattering matrix does not couple electrons and holes, we can consider separately the electron block $s_e$ and the hole block $s_h(t, t') = s_e^*(t, t')$. The current is then given by

$$
I(t) = -\frac{e}{2\pi} \int_{-\infty}^{\infty} dt' \int_{-\infty}^{\infty} dt'' \frac{1}{t'' - t'} \operatorname{Im} \operatorname{Tr} s_e^\dagger(t, t') P_{\mathrm{out}} s_e(t, t''). \tag{B.13}
$$

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
