# Peer review of "Time-resolved electrical detection of chiral edge vortex braiding"

_SciPost Physics_

## Round 2 · Referee Report · Fnu Setiawan · 2019-11-15

Strengths
1. Well written
2. Derivations of formulae are shown step by step
3. Clear objective
4. Results are clearly presented
Report
In the manuscript, the authors proposed a way to detect the non-Abelian braiding of chiral edge vortices via an electrical signal. They showed that the transfer of an electron from one vortex pair to another during the non-Abelian exchange can manifest as a flipping in the sign of the current pulse when the two vortices merge at the metal contact.
Compared to the previous studies Ref. [23,24], in this work, the author relax the assumption that the propagation time scale of the edge vortices is small compared to the injection time. The paper is very well written where the results and derivations of each formula are clearly shown.
I highly recommend this paper for a publication in Scipost as the result in this manuscript will significantly advance the field of topological quantum computation especially the braiding aspect of chiral edge vortices.
Requested changes
I suggest the authors to put explanations for each step of the braiding process in the caption of Figure 7.

---

## Round 2 · Referee Report · Anonymous · 2019-12-4

Report
In this work the authors propose a novel route to probing the non-Abelian exchange statistics of Majorana zero-modes, via a deterministic injection of edge vortices. To the best of my understanding, the main novel feature in this proposal as compared to the previous proposals in Refs. 23, 24, is the fact that multiple pairs of edge vortices are injected on demand and a braiding operation operation can be performed between vortices from different pairs. This allows for an extremely high level of control and therefore would be very desirable from an experimental point of view. Hence, I think this work could be an important contribution to the field, and would recommend its publication.
However, prior to publication, I would ask the authors to clarify a few points:
- The form of the effective Hamiltonian in Eq. (2.2), is the same as in Ref. 23, where an instantaneous scattering approximation was used. In Ref. 23 it is stated explicitly that the approximation is valid in the limit when the transit time through the system is short compared to the injection time. In the present work the authors work in the exactly opposite limit, hence it is not clear why this effective Hamiltonian can be used. In particular, in the braiding scheme presented in Fig. 5 - when a 2\pi flux is introduced across the second junction and a vortex is injected (in the second step shown in the figure) - is it clear that the vortices created previously - \sigma_1 and \sigma_2 are unaffected?
- Could the authors extend and clarify the comment in the last paragraph of the conclusions section: "Since only integer charge can enter into a normal metal, the fractional charge transfer by edge vortices cannot be noiseless"? What does this imply for the experiments?
- In Eq. (7.6) the authors give an expression for the total charge transferred, which is time independent but denote it by Q(t). I understand this to be a typo and that this should be simply Q.

---

## Editorial Decision

unknown